# Multi-Channel Signals in Dynamic Force-Clamp Mode of Microcantilever Sensors for Detecting Cellular Peripheral Brush

**DOI:** 10.3390/s24196312

**Published:** 2024-09-29

**Authors:** Qiang Lyu, Fan Pei, Ying-Long Zhao, Jia-Wei Ling, Neng-Hui Zhang

**Affiliations:** 1Shanghai Key Laboratory of Mechanics in Energy Engineering, Shanghai Institute of Applied Mathematics and Mechanics, School of Mechanics and Engineering Science, Shanghai University, Shanghai 200072, China; lvqiang@xynu.edu.cn (Q.L.); ljw0910@shu.edu.cn (J.-W.L.); 2College of Architecture and Civil Engineering, Henan New Environmentally-Friendly Civil Engineering Materials Engineering Research Center, Xinyang Normal University, Xinyang 464000, China; peifan2769@163.com (F.P.); yinglongzhao@yeah.net (Y.-L.Z.)

**Keywords:** peripheral brush, cellular viscoelasticity, microcantilever, detection mechanism, multi-channel signals

## Abstract

The development of numerous diseases, such as renal cyst, cancer, and viral infection, is closely associated with the pathological changes and defects in the cellular peripheral brush. Therefore, it is necessary to develop a potential new method to detect lesions of cellular peripheral brush. Here, a piecewise linear viscoelastic constitutive model of cell is established considering the joint contribution of the peripheral brush and intra-cellular structure. By combining the Laplace transformation and its inverse transformation, and the differential method in the temporal domain and differential quadrature method (DQM) in the spatial domain, the signal interpretation models for quasi-static and dynamic signals of microcantilever are solved. The influence mechanisms of the peripheral brush on the viscoelastic properties of cells and quasi-static/dynamic signals of microcantilever are clarified. The results not only reveal that the peripheral brush has significant effects on the complex modulus of the cell and multi-channel signals of the microcantilever, but also suggest that an alternative mapping method by collecting multi-channel signals including quasi-static and higher frequency signals with more brush indexes could be potentially used to identify cancerous cells.

## 1. Introduction

The cellular brush is one of the important peripheral microstructures in a living cell, which consist mainly of microvilli, microridges, and cilia [1], and plays an important role in physiological processes such as cell–cell interaction, cell migration, differentiation and proliferation, embryonic development, inflammation, and tumors [2,3]. Relevant experiments based on atomic force microscopy (AFM) have shown that the cellular peripheral brush can affect the mechanical responses of cells, thus changing the static/dynamic detection signals of AFM microscale sensors [4,5,6]. Thus, it is imperative to develop a reasonable constitutive model to accurately reflect the cellular peripheral brush and to take the mechanical properties of the peripheral brush as crucial indicators to distinguish between normal and cancerous cells. This exploration will significantly contribute to early cancer detection and treatment.

In the theoretical and experimental studies of cell mechanics, the interpretation of experimental results is predominantly based on the continuum medium model, in which cells are typically considered as homogeneous elastic or viscoelastic materials [7,8,9,10]. The AFM-based detection techniques for the mechanical properties of cells can be categorized into quasi-static mode and dynamic mode [11]. The AFM quasi-static mode mainly collects the signal of the force–distance curve and mainly includes force volume (FV), fast force volume (FFV), and fast force mapping modes [11,12,13,14,15]. The AFM dynamic mode mainly collects the signals of tip–sample interaction force, amplitude and phase [11], and mainly includes the multi-harmonic mode [16], sub-resonance mode [17], and AFM dynamic force-clamp mode [18,19,20,21]. However, with the change in AFM detection mode, there exists significant variations in the measured values of cellular apparent elastic modulus, thus indicating that a pure elastic model cannot fully capture proactive cellular behavior. Consequently, viscoelastic models have emerged as a preferred choice for investigating mechanical properties of cells [22,23,24,25], among which simpler ones include Maxwell and Kelvin–Voigt models. Nevertheless, these models often fall short in quantitatively describing complex viscoelastic behaviors exhibited by cells. In recent years, the standard linear solid model and power law model have been extensively utilized by numerous scholars to effectively characterize the viscoelastic behavior of cells, serving as single-domain constitutive models for cells [21,22,24].

The single-domain constitutive model of cells has been inadequate in accurately describing the dynamic changes in cellular peripheral brush bristles and other microstructures following cytopathic alterations and their effects on the multi-channel signals of AFM microbeams. Xu et al. established a self-similar multilevel structural model of cells consisting of the membrane, cytoskeleton, and cytoplasm [10]. In AFM peakforce tapping mode (dynamic mode), Schillers et al. demonstrated that the microvilli of MDCK C11 cells can be clearly scanned by controlling force in the pico-Newton range (100–200 pN). The length and diameter of the peripheral brush are approximately 1 μm and 200 nm, respectively [26]. In AFM indentation experiments (quasi-static mode), Iyer et al. and Sokolov et al. employed the single/double brush models to investigate the mechanical properties of MCF-7 cellular peripheral brush. They found that the reduction in stiffness after cell canceration is likely not due to the changes within the cell itself but rather derived from the peripheral brush. This is because experiments have shown that the physical parameters such as density and length of the peripheral brush are changed significantly after cell becomes cancerous [1,27]. The peripheral brush is generally regarded as a nonlinear elastic material, typically described by using the freely jointed chain (FJC) and worm-like chain (WLC) models of polymers and macromolecules [28]. In summary, the single-domain constitutive relationship of cells is insufficient to describe the dynamic adjustments in cell structures, viscoelastic properties, and functions. The peripheral brush effect should be considered when establishing a cell constitutive model.

This study focuses on a potential method to detect the effect of cancerous brush on cellular viscoelasticity by using multi-channel signals in dynamic force-clamp mode of microcantilever sensors. Firstly, based on the experimental discovery of the cellular peripheral brush [1,27], a piecewise linear viscoelastic constitutive model of the cell is established considering the nonlinear elasticity of the peripheral brush and the viscoelasticity of the intra-cellular structure. Secondly, the signal interpretation models for the quasi-static and dynamic signals of microcantilever system are solved by combining the Laplace transformation and its inverse transformation, and the differential method in the temporal domain and differential quadrature method (DQM) in the spatial domain. Finally, the influence mechanisms of the peripheral brush on the quasi-static and dynamic signals of the AFM microcantilever system are clarified. Also, based on AFM dynamic force-clamp mode, an alternative mapping method with multiple indicators including quasi-static signal and higher frequency peak amplitude is proposed to identify the cancerous cells.

## 2. Problem Formulation

As shown in Figure 1, the dynamic force-clamp mode is a potential AFM-based detection method, which is different from AFM tapping mode [20,21]. There are two piezo drivers in the detection system. The distance between the cell and the free-end of the microcantilever is changed by pulling or pushing the right-hand piezo driver to make the cell strained or compressed while a small-amplitude dithering is simultaneously applied to the fixed-end of microcantilever by the left-hand piezo driver with a low-driving frequency (i.e., 40–240 Hz). So, the multi-channel signals of the interaction force (static force signal) between the microcantilever tip and cell sample, the amplitude and phase of microcantilever motion can be collected simultaneously [20]. The length of the microcantilever is *l*, and the origin of coordinate *z* is located at the position where the interaction force between the fixed cell and microcantilever tip is zero, i.e., *F*_0_ = 0. A small periodic excitation displacement at the left-hand fixed-end is *w*(0) = *A*sin*ωt*, and the displacement and movement speed of the right-hand piezo driver are *z*(*t*) and *v*(*t*), respectively. The above two kinds of excitations induce the microcantilever–cell system, generating the following responses: *w*(*x*, *t*) and *F*(*t*), where *w*(*x*, *t*) is the bending deflection of microcantilever in *z* direction and *F*(*t*) is the interaction force between the deformed cell and microcantilever tip. The total deformation of cell is Δd, Δd=z(t)−w=vt−w.

Previous experimental studies have shown that the peripheral brush is composed of membrane protrusions, such as microvilli, microridges, cilia, and molecules grafted to the cellular membrane, such as glycosaccharide and glycoproteins (glycocalyx) [27]. In some SEM and EM scanning experiments, there were a lot of microvilli on the cell surface [1], which was regarded as a columnar structure, and included the microfilaments [29]. The molecules, microvilli, microridges, and cilia are usually characterized by the single/double brush model, elastic/viscoelastic model, freely jointed chain model (FJC), and worm chain model (WLC) [1,17,20,28]. The cell thickness is about 2–15 μm [30]. The length of the peripheral brush layer is 0.1–5 μm, which is 1/10–1/3 of the overall length of cell [1,27,31]. Based on the above research, as shown in Figure 2, the peripheral brush layer is coarsely grained into a given number of springs (disperse model), and the single brush is assumed as a nonlinear elastic spring, while the intra-cellular structure is simplified as a continuous viscoelastic cylinder (standard linear solid model) [21,30,32].

### 2.1. Two-Domain Structure Model

Previous experimental studies have shown that the mechanical behavior of a single brush can be represented by a two-stage linear elastic constitutive model (soft spring and hard spring) based on the force–displacement curve of single biomolecule experiments [28]. To enhance computational efficiency and applicability of cellular constitutive model, the piecewise linearization method is adopted to characterize the nonlinear elastic of single cellular brush [33].

When the strain of single brush is less than the critical strain, the brush is in the stage 1, i.e., tensile/compressive state (*i* = 1); otherwise, the brush is in the stage 2, i.e., tensile/compressive state (*i* = 2). Thus, the piecewise linear constitutive relationship of single brush can be expressed as
(1)$$f_{b}=f_{i}C=Ck_{bi}\delta_{b}=Ck_{bi}l_{b}\varepsilon_{b}=E_{bi}\varepsilon_{\mathrm{b.}}(i=1,\;2)$$
where *f*_b_ and *f_i_* represent the force per unit area in brush layer and the force of single brush, respectively; *C* is the brush density; *k*_b*i*_ and *E*_b*i*_ are the segmental stiffness of single brush and the elastic modulus of brush layer, respectively; and $\delta_{b}$, $\varepsilon_{b}$, and $l_{b}$ are the tensile/compressive displacement, the strain, and the length of single brush, respectively.

For the intra-cellular structure, a differential type viscoelastic constitutive relationship is expressed by the following standard linear solid model [21,30,32]:(2)$$(E_{1}+E_{2})\sigma +\eta_{1}\dot{\sigma }=E_{1}E_{2}\varepsilon_{v}+E_{2}\eta_{1}{\dot{\varepsilon }}_{v},$$
where *E*_1_, *E*_2_, and $\eta_{1}$ are the elastic and viscous parameters of the intra-cellular structure and σ and $\varepsilon_{v}$ are the stress and strain of the intra-cellular structure. The corresponding integral type viscoelastic constitutive relation can be written as
(3)$$\sigma =E_{v}⊗\varepsilon_{v}$$
where *E*_v_ is the relaxation function of the intra-cellular structure and $E_{v}=e_{0}+e_{1}\exp(-\alpha t)$, in which *e*_0_, *e*_1_, and *α* are the material parameters of the intra-cellular structure, which can be represented by *E*_1_, *E*_2_, and $\eta_{1}$, i.e., $e_{0}=E_{1}E_{2}/(E_{1}+E_{2})$, $e_{1}=E_{2}^{2}/(E_{1}+E_{2})$, $\alpha =(E_{1}+E_{2})/\eta_{1}$. The symbol ⊗ expresses the following linear Boltzmann operator [32]: $\psi_{1}(t)⊗\psi_{2}(t)=$ $\psi_{1}(0)\psi_{2}(t)+\int_{0}^{t}{\dot{\psi }}_{1}(t-\tau )\psi_{2}(\tau )d\tau$, where the superscript “⋅” represents the derivative of time *t*.

Peripheral brush layer is treated as a dispersed material which consists of a certain number of single brushes. Hence, we take into account that the force in brush layer needs to be fully transmitted to the cellular inner region:(4)$$f_{b}\times A_{b}=f_{b}\times A_{v}\times \lambda =\sigma \times A_{v}$$
where *A*_v_ is the contact area between peripheral brush and intra-cellular structure; *A*_b_ is the cross-section area of brush layer; and λ is the contact area ratio of brush layer to the cellular inner region, i.e., $\lambda =A_{b}/A_{v}$, which is the constant value and newly defined parameter resulting from the simplification of the intra-cellular structure into a viscoelastic cylinder and peripheral brush layer into a given number of springs. In addition, the total strain of cell is
(5)$$\varepsilon =\frac{\Delta d}{l_{b}+l_{v}}=\frac{\delta_{b}+\delta_{v}}{l_{b}+l_{v}}=\frac{l_{b}}{l_{b}+l_{v}}\varepsilon_{b}+\frac{l_{v}}{l_{b}+l_{v}}\varepsilon_{v},$$
where $\delta_{b}$ and $\delta_{v}$ are the tensile/compressive displacements of the peripheral brush and intra-cellular structure, respectively. *l*_v_ is the length of the intra-cellular structure. The total cellular deformation is given as $\Delta d=\delta_{b}+\delta_{v}=z(t)-w$. By applying the inverse Laplace transformation to Equations (1), (2), (4) and (5), we can obtain the following constitutive model of cell:(6)$$(E_{1}+E_{2})\bar{\sigma }+\eta_{1}\bar{\sigma }s=E_{1}E_{2}\left[\frac{\left(l_{b}+l_{v}\right)}{l_{v}}\bar{\varepsilon }-\frac{\bar{\sigma }}{\lambda E_{bi}}\right]+E_{2}\eta_{1}\left[\frac{\left(l_{b}+l_{v}\right)}{l_{v}}\bar{\varepsilon }-\frac{\bar{\sigma }}{\lambda E_{bi}}\right]s.$$

Applying the inverse Laplace transformation to Equation (6), the differential constitutive model of cell in the temporal domain can be obtained as
(7)$$\sigma +p_{1i}\dot{\sigma }=q_{0i}\varepsilon +q_{1i}\dot{\varepsilon },$$
where the differential operators *p*_1*i*_, *q*_0*i*_ and *q*_1*i*_ are expressed as
(8)$$\begin{array}{l} p_{1i}=\frac{(\lambda E_{bi}+E_{2})\eta_{1}}{\lambda E_{bi}(E_{1}+E_{2})+E_{1}E_{2}},\;\\ q_{0i}=\frac{\lambda E_{bi}E_{1}E_{2}(l_{v}+l_{b})}{[\lambda E_{bi}(E_{1}+E_{2})+E_{1}E_{2}]l_{v}},\;\\ q_{1i}=\frac{\lambda E_{bi}E_{2}\eta_{1}(l_{v}+l_{b})}{[\lambda E_{bi}(E_{1}+E_{2})+E_{1}E_{2}]l_{v}}. \end{array}$$

Note that the contribution of both the peripheral brush and intra-cellular structure is considered in the present differential constitutive model. When there is no peripheral brush (i.e., when $l_{b}=0$), Equation (7) can be reduced to the standard linear solid model [21,30,32], which demonstrates the compatibility of present model, validating the above derivation. The corresponding integral type of Equation (7) can be similarly expressed as
(9)$$\sigma =E_{i}^{*}⊗\varepsilon ,$$
where $E_{i}^{*}$ is the relaxation function of the stretched/compressed cell, which is expressed as
(10)$$\begin{array}{l} E_{i}^{*}=e_{0i}+e_{1i}e^{-\alpha_{i}t},\\ e_{0i}=q_{0i},e_{1i}=q_{1i}/p_{1i}-q_{0i},\alpha_{i}=1/p_{1i}. \end{array}$$

When stress and strain are oscillatory functions, let s=iω, then the differential operators P(iω) and Q(iω) in the frequency domain can be expressed as
(11)$$P(i\omega )=1+p_{1i}i\omega ,\;Q(i\omega )=q_{0i}+q_{1i}i\omega .$$

The complex modulus of cell in the frequency domain can be expressed as
(12a)$$E^{*}(i\omega )=\frac{Q(i\omega )}{P(i\omega )}=\frac{q_{0i}+q_{1i}i\omega }{1+p_{1i}i\omega },$$
(12b)$$E_{1}^{*}(\omega )=\frac{q_{0i}+p_{1i}q_{1i}\omega^{2}}{1+{\left(p_{1i}\omega \right)}^{2}},\;E_{2}^{*}(\omega )=\frac{(q_{1i}-p_{1i}q_{0i})\omega }{1+{\left(p_{1i}\omega \right)}^{2}},$$
where $E_{1}^{*}(\omega )$ and $E_{2}^{*}(\omega )$ are the energy storage modulus and loss modulus of cell, respectively.

### 2.2. Interpretation Models of Microcantilever Signal

Here, referring to our previous work [21], a small periodic excitation *A*sin*ωt* is applied to the fixed-end in dynamic force-clamp detection mode, and the stretched/compressed cell deformation in *z* direction is $\Delta d(t)=z_{0}+vt-{\left.w(x,\;t)\right|}_{x=l}$. Thus, the governing equation for the dynamic response of microcantilever in the temporal domain can be written as
(13)$$m\ddot{w}(x,t)+\gamma \dot{w}(x,t)+EI\frac{\partial^{4}w(x,t)}{\partial x^{4}}=0,$$
where *m* is the mass per unit length of microcantilever and *γ* is the dissipative coefficient of background. The cell environment is in the air, and the related effects in the liquid environment are not considered in the model [21,34]. It should be noted that due to the contribution of the peripheral brush and intra-cellular structure in the present model, the boundary interaction force is characterized in different forms, so the boundary conditions are as follows:$${\left.w\right|}_{x=0}=A\sin\omega t,\ldots {\left.\frac{\partial w}{\partial x}\right|}_{x=0}=0,$$
(14)$${\left.\frac{\partial^{2}w}{\partial x^{2}}\right|}_{x=l}=0,\ldots \begin{array}{l} EI{\left.\frac{\partial^{3}w}{\partial x^{3}}\right|}_{x=l}=F(t)=-S\left[E_{i}^{*}⊗\Delta d/(l_{b}+l_{v})\right]\\ =-S\left[E_{i}^{*}⊗\left(z_{0}+vt-{\left.w\right|}_{x=l}\right)/\left(l_{b}+l_{v}\right)\right], \end{array}$$
where *S* is the contact area between the microcantilever tip and cell, and the contact area is assumed no change with time [21,35]; *z*_0_ is the initial position of substrate. For the AFM quasi-static mode, the above governing equation and boundary conditions can degenerate into
(15a)$$EI\frac{\partial^{4}w_{s}(x,t)}{\partial x^{4}}=0,$$
$${\left.w_{s}\right|}_{x=0}=0,\ldots {\left.\frac{\partial w_{s}}{\partial x}\right|}_{x=0}=0,$$
(15b)$${\left.\frac{\partial^{2}w_{s}}{\partial x^{2}}\right|}_{x=l}=0,\ldots EI{\left.\frac{\partial^{3}w_{s}}{\partial x^{3}}\right|}_{x=l}=F_{s}(t)=-S\left[E_{i}^{*}⊗\left(z_{0}-{\left.w\right|}_{x=l}\right)/l_{b}+l_{v}\right].$$

## 3. Solution Methods

In this section, several methods are combined to solve the above mathematical models for quasi-static/dynamic responses of the microcantilever. Firstly, by combining the Laplace transformation and its inverse transformation, an analytical solution for the quasi-static response of the microcantilever is obtained. Secondly, by the combining differential method [36] in the temporal domain and DQM [37] in the spatial domain, a numerical solution for the dynamic response of microcantilever is obtained.

In AFM quasi-static contact mode, applying the Laplace transformation to Equation (15) yields the transformed deflection in the Laplace domain as
(16)$${\bar{w}}_{s}=\left[S{\bar{E}}_{i}^{*}z_{0}s/\left(2Sl^{3}{\bar{E}}_{i}^{*}s^{2}+6EI(l_{b}+l_{v})\right)s\right](3lx^{2}-x^{3}),$$
where $\bar{\left[\;\right]}$ represents the Laplace transformation and *s* is the transformation parameter.

By applying the Laplace transformation to Equation (10), substituting the result into Equation (16), and then applying the inverse Laplace transformation to Equation (16), we can obtain the analytic solution for quasi-static deflection of microcantilever as
(17)$$w_{s}=a_{1}\left[a_{2}\exp(-\beta t)+a_{3}\right](3lx^{2}-x^{3}),$$
where $a_{1},\;a_{2},\;a_{3}$, β are the polynomial coefficients, and
(18)$$a_{1}=S{\left[Sl^{3}(e_{0i}+e_{1i})+3EI(l_{b}+l_{v})\right]}^{-1}{\left[Sl^{3}e_{0i}+3EI(l_{b}+l_{v})\right]}^{-1}/2,\;a_{2}=3EIe_{1}z_{0}(l_{b}+l_{v}),\;a_{3}=e_{0i}z_{0}\left[Sl^{3}(e_{0i}+e_{1i})+3EI(l_{b}+l_{v})\right],\;\beta =\frac{3EI(l_{b}+l_{v})+Sl^{3}e_{0i}}{3EI(l_{b}+l_{v})+Sl^{3}(e_{0i}+e_{1i})}\alpha .$$

In AFM dynamic force-clamp mode, the differential method in the temporal domain and DQM in the spatial domain are combined to solve the governing equation for dynamic response of the microcantilever. The length of microcantilever is discretized into *N* points in *x* direction. The first-order derivative of function *w* at the given discrete point *x* = *x_i_* for the independent variable *x* can be expressed as the following matrix form:(19)$${w^{\prime }}_{i}={\left(\frac{\partial w}{\partial x}\right)}_{i}=\sum_{j=1}^{N}A_{ij}w_{j}=A_{ij}w_{j},\;i,\;j=1,\;2,\;\cdots \cdots ,\;N$$
where *w_j_* is the desired *N* line vector of function *w* and ***A****_ij_* is the weighted coefficient matrix of the first-order partial derivative, which is obtained by using the Lagrange interpolation polynomial. The second-, third-, and fourth-order derivatives can be conveniently obtained by the product of lower weight coefficient matrices, i.e.,
(20)$$\begin{array}{l} {w^{\prime \prime }}_{i}=A_{ij}{w^{\prime }}_{j}=B_{ij}w_{j},\\ {w^{\prime \prime \prime }}_{i}=A_{ij}{w^{\prime \prime }}_{j}=C_{ij}w_{j},\\ w_{i}^{\left(4\right)}=B_{ij}{w^{\prime \prime }}_{ij}=D_{ij}w_{j}. \end{array}$$

In this paper, the well-accepted Gauss–Chebyshev–Lobatto-type distribution of grid point is adopted as [38]
(21)$$x_{i}=\frac{l}{2}\left[1-\cos\frac{\left(i-1\right)\pi }{N-1}\right],\;i=1,\;2,\;\cdots \cdots ,\;N,$$
where *x_i_* is the spacing grids in *x* direction.

By using Equations (19)–(21), we can transform Equation (13) into
(22)$$m{\ddot{w}}_{j}+\gamma {\dot{w}}_{j}+EID_{ij}w_{j}=0.$$

Similarly, Equation (14) can be discretized as
(23)$$w_{1j}=A\sin\omega t,\;A_{1j}w_{j}=0,\;B_{Nj}w_{j}=0,\;[EI(l_{b}+l_{v})/S]C_{Nj}({\dot{w}}_{j}+\alpha w_{j})-E_{i}^{*}(0)(\Delta {\dot{d}}_{j}+\alpha_{i}\Delta d_{j})+\alpha_{i}e_{1}\Delta d_{j}=0.$$

The initial conditions the dynamic response of microcantilever are written as
(24)$$w_{j}{|}_{t=0}=w_{j}^{0},\;{\dot{w}}_{j}{|}_{t=0}={\dot{w}}_{j}^{0}$$
where $w_{j}^{0}$, ${\dot{w}}_{j}^{0}$ can be obtained in experiments. The numerical solution for dynamic response can be obtained by programming Equations (22)–(24) with the Matlab R2018b.

## 4. Results and Discussion

In this section, the effects of the elastic properties of the peripheral brush on the cell viscoelastic properties and the quasi-static/dynamic responses of the microcantilever are explored. Firstly, in order to verify the effectiveness of present model, the present analytical result of quasi-static response is compared with the relevant experiment [28,30], and the convergence of DQM is verified in AFM dynamic force-clamp mode. Secondly, the influence of the peripheral brush parameters on cell viscoelastic properties is discussed. Finally, the effects of the peripheral brush on the quasi-static and dynamic responses are predicted. Therefore, an alternative mapping method is proposed to identify the cancerous cells in AFM dynamic force-clamp mode. In computation, a photoresistant SU-8 microcantilever is taken with the geometric parameters of *l* = 200 μm, *b* = 20 μm, *h*_s_ = 0.8 μm, and S = π(300)^2^ nm^2^ and the material parameters of *E* = 4.0 GPa, *ρ* = 1190 kg/m^3^ [21,23].

### 4.1. Model Verification

The present analytical result is compared with the experimental results of Chtcheglova’s et al. by using Equation (15) [28] in the AFM tensile experiment (quasi-static contact mode). As shown in Figure 3a, the segmental stiffness of the peripheral brush are taken as *k*_b1_ = 0.00078 N/m and *k*_b2_ = 0.0125 N/m, which are consistent with the range of elastic constants (0.0001–0.025 N/m) of a single protein molecule determined by Chtcheglova et al. [28]. The coefficients of determination (*R*^2^) for the piecewise curve fittings based on the present model are 0.944 and 0.982. In the AFM relaxation experiment (quasi-static contact mode), the present analytical result based on Equation (17) is compared with the experimental results of Darling’s et al. [30]. As shown in Figure 3b, when the peripheral brush is considered, the determination coefficient for the curve fitting based on the present model is 0.958; while, when the peripheral brush is not considered (SLS model), the determination coefficient for the curve fitting is 0.927 [21]. The fitted cellular parameters are listed in Table 1. However, from Table 1, the mechanical parameters fitted by the present model and its reduced model (no brush) [21] are significantly different from those obtained by the Darling’s model due to different theories and cell constitutive models. It indicates that the present model, which considers the cellular microstructures (i.e., peripheral brush and intra-cellular structure), can better fit the relaxation experiment data. In AFM dynamic force-clamp mode, the amplitude–frequency curve for the dynamic response of the microcantilever is obtained by using Equations (22)–(24). The previous studies suggest that a low frequency excitation with small excitation amplitude seems eminently suitable for detecting biomolecular viscoelasticity under AFM dynamic force-clamp mode [21], and this frequency is not only higher than the resolutions of the detection frequency (i.e., 20 Hz) and amplitude (i.e., 0.2 nm), but also far from the thermal noise frequency (i.e., 500 Hz). As shown in Figure 3c, the numerical result converges rapidly with the increase in discretized point number *N* by using Gauss–Chebyshev–Lobatto–type distribution; when *N* = 9, the numerical results converge. The good consistencies demonstrate the convergence of DQM, and this validates the present method.

### 4.2. Peripheral Brush Effect on Cell Viscoelasticity

The influence of brush density, segmental stiffness, and brush length on the complex modulus of cell in the frequency domain will be discussed by using Equation (12). In calculation, the viscoelastic parameters of the intra-cellular structure of JJ012 cells are as follows: *e*_0_ = 230 Pa, *e*_1_ = 210 Pa, and *α* = 0.125 s^−1^ [30]. As shown in Figure 4, the relaxation modulus of cell in the frequency domain exhibits a decreasing trend after taking into account the elastic property of the peripheral brush. These results are be caused by the coupling effect between the elasticity of the peripheral brush and the viscoelasticity of the intra-cellular structure, as shown in Equations (8) and (12).

In addition, as seen in Figure 4a–c, with the increasing brush density, not only the segmental stiffness and brush length, but also the transient- and steady-state complex moduli of the cell in the frequency domain increase with the trend becoming closer to the complex modulus of the intra-cellular structure. For example, the modulus of cell is closer to that of the peripheral brush domain for the lower elastic modulus of the peripheral brush, whereas it is closer to that of the intra-cellular structure for the higher elastic modulus of the peripheral brush. Furthermore, the effect of the peripheral brush on the complex moduli of normal and cancerous cells in the frequency domain will be discussed. For human cervical epithelial cells, a normal cell has a brush length of ~2.4 μm with a grafting density of ~300 μm^−2^; however, the cancerous cell has two kinds of brushes with characteristic lengths of 0.45 and 2.6 μm and grafting densities ~640 and 180 μm^−2^, respectively [1]. As seen in Figure 4d, when the cell becomes cancerous, the brush number decreases significantly, which results in a decrease in the cellular complex modulus. These results also imply that we should carefully assert the previous conclusions based on those single domain constitutive models [30].

### 4.3. Peripheral Brush Effect on Quasi-Static Response of Microcantilever

In AFM quasi-static mode (AFM relaxation experiment), the initial static displacement of substrate is fixed (i.e., *z*_0_ = 200 nm). The influence of the brush density, segmental stiffness, and brush length on microcantilever quasi-static responses (i.e., the time–history curves of the deflection and interaction force) will be investigated by using Equation (17). In calculations, the viscoelastic parameters of the intra-cellular structure of JJ012 cells are as follows: *e*_0_ = 230 Pa, *e*_1_ = 210 Pa, and *α* = 0.125 s^−1^ [30]. Substituting Equation (17) into Equation (15b) yields the interaction force between the tip and cell, i.e., $F_{s}(t)=6EIa_{1}\left[a_{2}\exp(-\beta t)+a_{3}\right]$. As shown in Figure 5a,c,e, the quasi-static deflection of the microcantilever decreases when the peripheral brush is considered, whereas it increases with the increase in brush density, segmental stiffness, and brush length. This is due to the obvious effect of the peripheral brush on the overall mechanical properties of the cell, which leads to the decrease or increase in the interaction force between the microcantilever and the fixed cell, as shown in Figure 5b,d,f. Furthermore, as we can see in Figure 5g,h, for the human cervical epithelial cells, when the cell becomes cancerous, the density of the peripheral brush decreases significantly, which results in a decrease in the interaction force and quasi-static deflection. The above results show that the change in the peripheral brush has a significant effect on the quasi-static signals of the microcantilever, which can be used to identify the physical parameters of the peripheral brush, which is helpful for distinguishing the normal cells and cancerous cells.

### 4.4. Peripheral Brush Effect on Dynamic Response of Microcantilever

In AFM dynamic force-clamp mode, the influence of brush density, segmental stiffness, and brush length on the dynamic responses of the microcantilever are investigated by using Equations (22)–(24). With the movement of the pulling piezo driver (*v* = 25 nm/s) and a small periodic excitation of the dithering piezo driver (*A* = 10 nm, the frequency *f* = 100 Hz), the dynamic signals of the bending deflection–time curves of the microcantilever are obtained, as shown in the inset of Figure 6a–d. Figure 6a–d show the quasi-static signals of microcantilever caused by the movement of the pulling piezo driver (i.e., the force–displacement curve of the microcantilever without considering the contribution of small periodic excitation of the dithering piezo driver, *v* = 25 nm/s, *A*= 0 nm, *f* = 0 Hz). In Figure 6e–h, by using the fast Fourier transform method, the amplitude–frequency curve is generated from the dynamic bending deflection–time curve of the microcantilever, which is obtained by subtracting the quasi-static signals from the dynamic signals of the microcantilever. Note that the studied frequency is higher than the resolutions of the detection frequency (i.e., 20 Hz) and amplitude (i.e., 0.2 nm), but is also far from thermal noise frequency (i.e., 500 Hz). In calculations, the viscoelastic parameters of the intra-cellular structure of JJ012 cells are as follows: *e*_0_ = 230 Pa, *e*_1_ = 210 Pa, and *α* = 0.125 s^−1^ [30].

As shown in Figure 6a–d, the interaction force becomes stronger with the increase in the brush density, segmental stiffness, and brush length; when the cell becomes cancerous, the density of the peripheral brush decreased significantly, and these changes induce a decrease in interaction force. The results show that the change in the peripheral brush has a significant effect on the interaction force in AFM dynamic force-clamp mode, which could be used to identify the physical parameters of the peripheral brush, and this may be helpful for distinguishing the normal cells and cancerous cells. As shown in the inserts of Figure 6e–g, within the frequency (i.e., *f*) range of 0–200 Hz, there are two peaks (i.e., double-peak resonance) in the amplitude–frequency curve: one peak lies in the lower frequency domain, and the other lies in the higher frequency domain. This reminds us of the double-peak resonance phenomena for cell detection in our previous work [21]. The similar point of the previous work in Ref. [21] and the present study is as follows: multiple signal peaks appear in the amplitude–frequency curves for the same vibration mode. However, the interests are different. With the increasing brush density, segmental stiffness, and brush length, the higher frequency peak amplitude increases prominently, whereas the lower frequency peak amplitude and the interval between two peaks seem unchanged. In fact, this is because with the increasing brush density, segmental stiffness, and brush length, the increase in cell modulus at the higher frequency is more remarkable than that at the lower frequency, as shown in Figure 4. Furthermore, as shown in Figure 6h, the amplitude of the higher frequency peak decreases with the canceration of the cell, which is due to the changes in the density and length of the peripheral brush.

These results not only suggest that the quasi-static signal of microcantilever is an additional channel for detecting the brush density, segmental stiffness, and brush length in AFM quasi-static and dynamic force-clamp modes, but also point out that the higher frequency peak amplitude of the microcantilever can be used to detect the peripheral brush in AFM dynamic force-clamp mode. The effective extraction and rational use of multi-channel signals, including more microstructure information, may be beneficial for the early diagnosis of cancer.

### 4.5. Limitations of Present Model and Future Developments

Note that the present model is established for AFM tension/compression (indentation) states in air. The effects of the peripheral brush on the complex moduli and static and dynamic signals of the microcantilever are mainly discussed in the state of indentation compression. The contact part between the cell and the microcantilever is characterized as a viscoelastic moving boundary condition, which mainly focuses on the joint contribution of the peripheral brush and intra-cellular structure. The influence of the tip shape is ignored and the contact area is assumed to be unchanged with time. In the future, a more completed model for the cellular peripheral brush will be further investigated, and the fluid–solid coupling model in consideration of liquid effects and the tip shape of microcantilever will also be further investigated.

## 5. Conclusions

Based on the experimental findings related to the cellular peripheral brush [27], a piecewise linear viscoelastic constitutive model of cell is established considering the joint contribution of the peripheral brush and intra-cellular structure. We clarify the influence mechanism of the peripheral brush effect on the viscoelastic properties of the cell and the multi-channel mechanical signals of the microcantilever. Therefore, an alternative potential detection method with multi-channel signals including quasi-static and higher frequency signals of microcantilever is proposed. The present results agree well with the previous studies in AFM quasi-static contact mode and AFM dynamic force-clamp mode. The main conclusions are as follows:(1)The complex modulus of the cell in the frequency domain, the quasi-static deflection/interaction force, and the higher frequency peak amplitude of the microcantilever enhance with the increase in brush density, segmental stiffness, and brush length. These results indicate the necessity of considering the influence of the peripheral brush in the establishment of the interpretation model of the microcantilever signal.(2)In AFM dynamic force-clamp mode, the quasi-static deflection/interaction force and higher frequency peak amplitude in double-peak resonance mode of the microcantilever can be used to obtain more indexes including brush density, segmental stiffness, and brush length for the diagnosis of cancerous cells.(3)However, the relevant AFM dynamic force-clamp experiments need to be further explored to verify the utilization of multi-channel signals with more indexes to identify cancerous cells.

## Figures and Tables

**Figure 1 sensors-24-06312-f001:**
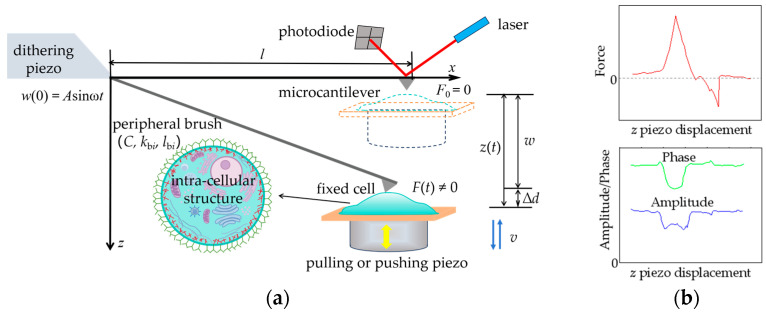
Schematic diagrams of protocol. (**a**) Microcantilever deflection and cellular microstructure in AFM dynamic force-clamp mode; (**b**) multi-channel signals of interaction force (static force signal) between microcantilever tip and cell sample, amplitude and phase of microcantilever.

**Figure 2 sensors-24-06312-f002:**
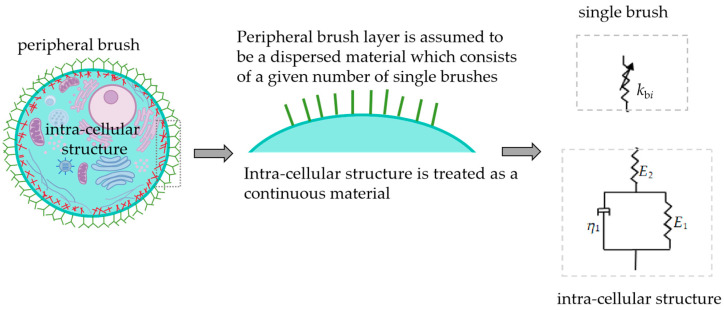
Schematic diagram of cellular microstructure and its constitutive model.

**Figure 3 sensors-24-06312-f003:**
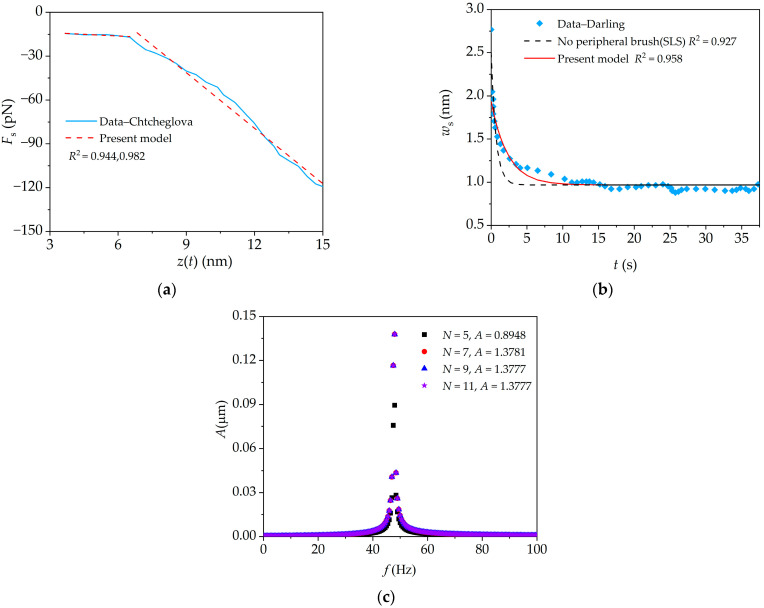
Verification of present model. (**a**) Force–displacement curve in AFM quasi-static tensile experiment; (**b**) time–history curve for bending deflection of microcantilever in AFM quasi-static relaxation experiment (*z*_0_ = 50 nm); (**c**) amplitude–frequency curve of microcantilever in AFM dynamic force-clamp mode.

**Figure 4 sensors-24-06312-f004:**
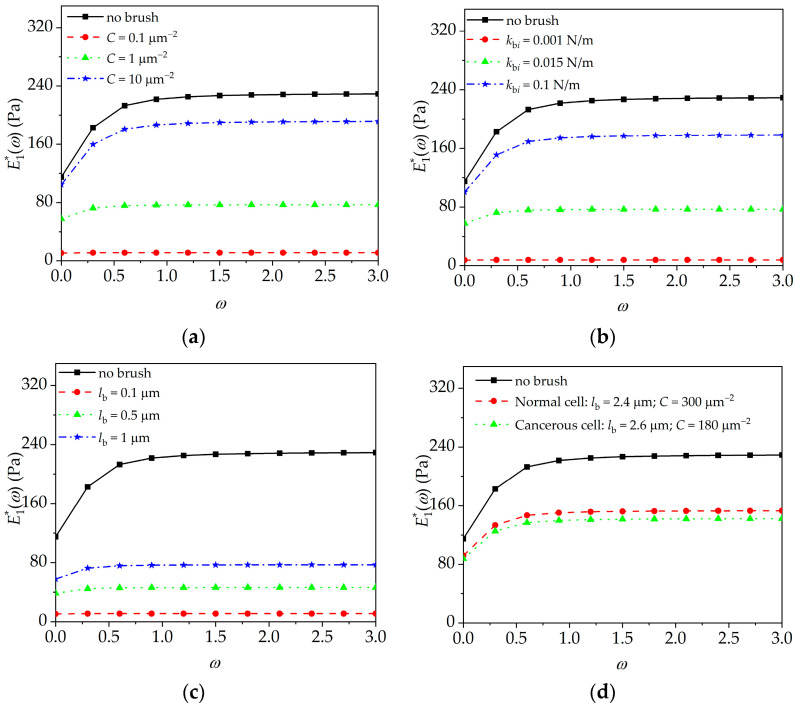
Peripheral brush effect on the energy storage modulus of normal and cancerous cells. (**a**) Brush density (*l*_b*i*_ = 1 μm, *k*_b*i*_ = 0.015 N/m); (**b**) segmental stiffness (*l*_b*i*_ = 1 μm, *C* = 10 μm^−2^); (**c**) brush length (*k*_b*i*_ = 0.015 N/m, *C* = 10 μm^−2^); (**d**) peripheral brushes of normal and cancerous cells (*k*_b*i*_ = 0.01 N/m).

**Figure 5 sensors-24-06312-f005:**
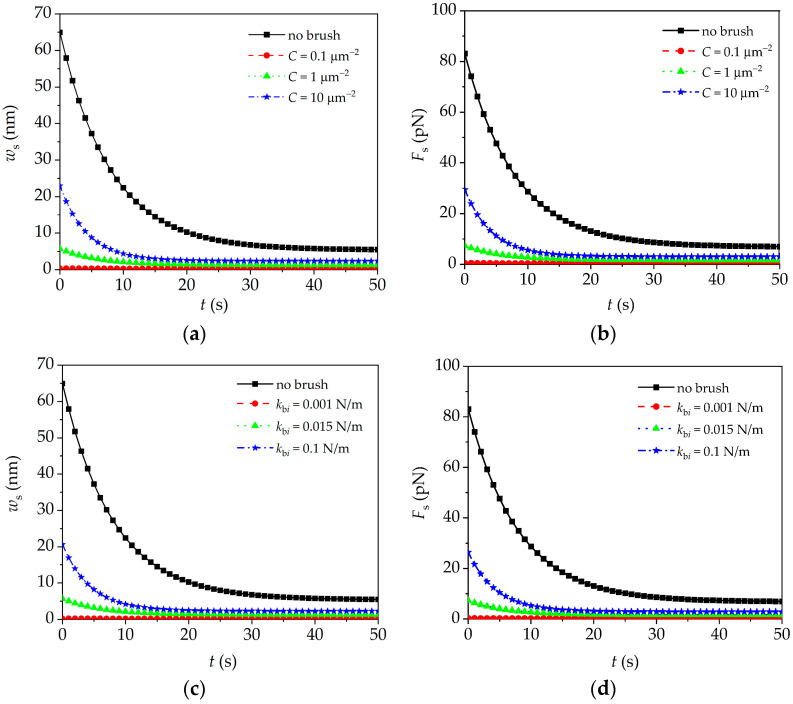
Peripheral brush effect on quasi-static signals of microcantilever. (**a**,**b**) Brush density (*l*_b*i*_ = 1 μm, *k*_b*i*_ = 0.015 N/m); (**c**,**d**) segmental stiffness (*l*_b*i*_ = 1 μm, *C* = 10 μm^−2^); (**e**,**f**) brush length (*k*_b*i*_ = 0.015 N/m, *C* = 10 μm^−2^); (**g**,**h**) peripheral brushes of normal and cancerous cells (*k*_b*i*_ = 0.015 N/m).

**Figure 6 sensors-24-06312-f006:**
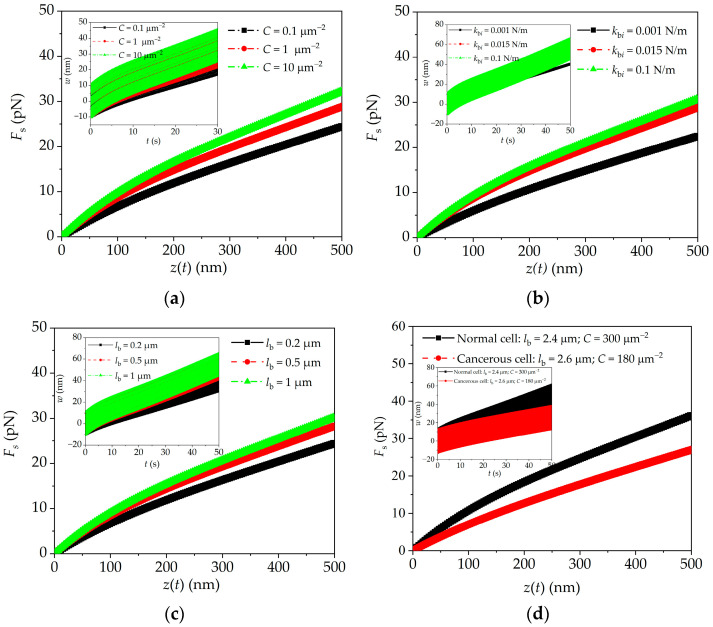
Peripheral brush effect on quasi-static and dynamic responses of microcantilever (the multi-channel signals caused by the movement of pulling piezo driver and small periodic excitation of dithering piezo driver are shown in the inset of Figure 6). (**a**–**c**) Brush density (*l*_b*i*_ = 1 μm, *k*_b*i*_ = 0.1 N/m), segmental stiffness (*l*_b*i*_ = 1 μm, *C* = 10 μm^−2^), and brush length (*k*_b*i*_ = 0.1 N/m, *C* = 10 μm^−2^) effects on quasi-static signals; (**d**) brush effects of normal and cancerous cells on quasi-static signals (*k*_b*i*_ = 0.1 N/m); (**e**–**g**) brush density (*l*_b*i*_ = 1 μm, *k*_b*i*_ = 0.1 N/m), segmental stiffness (*l*_b*i*_ = 1 μm, *C* = 10 μm^−2^), and brush length (*k*_b*i*_ = 0.1 N/m, *C* = 10 μm^−2^) effects on dynamic characteristics; (**h**) brush effect of normal and cancerous cells on dynamic characteristics (*k*_b*i*_ = 0.1 N/m).

**Table 1 sensors-24-06312-t001:** Mechanical parameters of JJ012 cells in AFM stress relaxation experiment.

Cellular Parameters	Darling [30]	SLS (No Brush) [21]	Present Model
*e*_0_*_i_* (Pa)	230 ± 150	26.33 ± 1.67 × 10^3^	185.65 ± 61.2
*e*_1*i*_ (Pa)	210 ± 130	41.59 ± 3.3 × 10^3^	21.05 ± 10.9
*e*_0_*_i_* _+_ *e*_1*i*_ (Pa)	440 ± 270		206.7 ± 72.1
*α_i_* (s^−1^)	0.125	1.491 ± 0.467	0.474
*E*_b*i*_ (Pa)			56.7 ± 3.8
*R* ^2^	0.875	0.927	0.958

## Data Availability

The data presented in this study are available on request from the corresponding author. The data are not publicly available due to privacy.

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
