# Peer review of "Multi-Channel Signals in Dynamic Force-Clamp Mode of Microcantilever Sensors for Detecting Cellular Peripheral Brush"

_sensors, 2024, doi:10.3390/s24196312_

Round 1

Reviewer 1 Report

Comments and Suggestions for Authors

The manuscript “Multi-Channel Signals in Dynamic Force-Clamp Mode of Microcantilever Sensors for Detecting Cellular Peripheral Brush” by Qiang Lyu et al. studies an influence of cellular peripheral brush on the measurement of cell viscoelastic properties in quasi-static/dynamic AFM experiments. A model of cell was established considering the joint contribution of peripheral brush and intra-cellular structure. The results indicate the necessity of considering the influence of peripheral brush (brush density, segmental stiffness, and brush length) in the interpretation of experimental AFM data.

The main idea of the manuscript is understandable, but there are many places where clarifications are needed. Therefore, I can’t recommend acceptance of the manuscript in the current form.

Comments:

1)    The used terminology, both in AFM and cell biology parts. Below are examples of questionable or wrong usage of the terms.

2)    Lines 42-45: “The primary AFM experiments utilized for detecting cell mechanics properties currently encompass: AFM contact mode [11,12], AFM tapping mode [13], and AFM dynamic force-clamp mode” Contact mode is generally an imaging mode (description of the mode can be found in the Wikipedia, and in any current review on the AFM principles, e.g., doi: 10.1038/nnano.2017.45, cited in the text). What the author meant here is probably an indentation or force curve/force spectroscopy mode. Tapping mode is not quite used for the mechanical analysis, unless the authors can elaborate it. Quasi-static and dynamic modes should be mentioned here, but they can be found only later in the text. These methods are also extensively reviewed (10.1039/C9SM00711C; 10.1039/D0CS00318B), and the authors should use the established terminology.

3)    “Dynamic Force-Clamp Mode” please add a scheme of this particular mode (e.g. cantilever displacement and force vs time), in Fig. 1.

4)    Lines 47-48. “In AFM dynamic force-clamp mode, the elastic modulus of living cell is obtained from both the AFM force-displacement curve and single peak amplitude-frequency curve [15,23].” It is not clear that is a single peak amplitude-frequency curve, the cited references are not informative about it too.

5)    Lines 63-65: “In AFM tapping mode, Schillers et al. demonstrated that the peripheral brush of MDCK C11 cells can be clearly scanned by controlling force in the pico-Newton range (100−200 pN).” This work is about microvilli and not about the peripheral brush. Brush is considered to be a layer of polysaccharides and glycoproteins attached to the plasma membrane. Please clarify that is the peripheral brush in the current work.

6)    Lines 73-74, “free link chain (FJC)” is it a correct acronym?

7)    Line 112, “diseased peripheral brushes are about 0.4 μm and 2.6 μm” – please clarify specific cells and that there were two types of brush observed in that study.

8)    In some of the cited studies (Chtcheglova et al. 2004) the mechanical parameter of brush is studied in tension experiments, while in the other ones – in compression. The properties of cells in AFM experiments are generally studied in compression (indentation) mode. Is there an assumption that properties of the brush are the same in tension and in compression? Is it a plausible assumption? That should be discussed. Also, in the model described by authors, the tension or compression experiments are supposed to happen?

9)    It is unclear which cantilever tip shape is considered in the developed model. Would be the conclusions different for cone, pyramid, cylinder, or microspheres of different radii?

10) Parameter λ of the model, the contact area ratio of peripheral brush to the cellular inner region, what was it? Would it depend on the tip shape? Is it a function of the strain?

11) Lines 191, 192 “γ is the dissipative coefficient of background (in air).” Why in air if all experiments with cells are performed in liquid?

12) When compared with the Darling’s results, could the authors obtain the same results when there is no peripheral brush in the model (reduced model)?

13) Please explain Fig. 3c, why the peak is around 50 Hz? How is it related to the Darling’s work?

14)  Figure 6. Too small text in subplots.

Comments on the Quality of English Language

Minor editing of English language required. Please use conventional AFM and biological terminology.

Author Response

Comments:

(1) The used terminology, both in AFM and cell biology parts. Below are examples of questionable or wrong usage of the terms.

Response: We agree with this comment. Therefore, we searched some literatures on AFM and cell biology and read them carefully. Then the relevant terms were explained and revised in detail in the manuscript.

(2) Lines 42-45: “The primary AFM experiments utilized for detecting cell mechanics properties currently encompass: AFM contact mode [11,12], AFM tapping mode [13], and AFM dynamic force-clamp mode” Contact mode is generally an imaging mode (description of the mode can be found in the Wikipedia, and in any current review on the AFM principles, e.g., doi: 10.1038/nnano.2017.45, cited in the text). What the author meant here is probably an indentation or force curve/force spectroscopy mode. Tapping mode is not quite used for the mechanical analysis, unless the authors can elaborate it. Quasi-static and dynamic modes should be mentioned here, but they can be found only later in the text. These methods are also extensively reviewed (10.1039/C9SM00711C; 10.1039/D0CS00318B), and the authors should use the established terminology.

Response: We agree with this comment. We have carefully read the AFM-handbook and some review articles, and we have rewritten the contents of lines 42-53 in manuscript. The established terminologies in AFM and cell biology parts were used.

(3) “Dynamic Force-Clamp Mode” please add a scheme of this particular mode (e.g. cantilever displacement and force vs time), in Fig. 1.

Response: We agree with this comment. We have added the scheme of cantilever displacement and force vs time in Fig. 1.

(4) Lines 47-48. “In AFM dynamic force-clamp mode, the elastic modulus of living cell is obtained from both the AFM force-displacement curve and single peak amplitude-frequency curve [15,23].” It is not clear that is a single peak amplitude-frequency curve, the cited references are not informative about it too.

Response: We agree with this comment. We have made a more accurate description of the relevant content in lines 42-53.

(5) Lines 63-65: “In AFM tapping mode, Schillers et al. demonstrated that the peripheral brush of MDCK C11 cells can be clearly scanned by controlling force in the pico-Newton range (100−200 pN).” This work is about microvilli and not about the peripheral brush. Brush is considered to be a layer of polysaccharides and glycoproteins attached to the plasma membrane. Please clarify that is the peripheral brush in the current work.

Response: Thanks for careful reading. We have added the consistent of peripheral brush (consist mainly of microvilli, microridges and cilia, as seen in Ref [1] in manuscript) in lines 31-32, and changed the “peripheral brush” to “microvilli” in lines 63-67.

(6) Lines 73-74, “free link chain (FJC)” is it a correct acronym?

Response: Thanks for careful reading. We have changed “free link chain (FJC)” to “free jointed chain (FJC)” in lines 76-77 and 122-123.

(7) Line 112, “diseased peripheral brushes are about 0.4 μm and 2.6 μm” – please clarify specific cells and that there were two types of brush observed in that study.

Response: Thanks for careful reading. We have added the specific cells and clarified the types of peripheral brush in lines 112-119.  

(8) In some of the cited studies (Chtcheglova et al. 2004) the mechanical parameter of brush is studied in tension experiments, while in the other ones – in compression. The properties of cells in AFM experiments are generally studied in compression (indentation) mode. Is there an assumption that properties of the brush are the same in tension and in compression? Is it a plausible assumption? That should be discussed. Also, in the model described by authors, the tension or compression experiments are supposed to happen?

Response: We agree with this comment. We have added the discussion in lines 392-395. The present model is established for AFM tension/compression (indentation) states. In the “Results and Discussion” section, the effects of peripheral brush on the complex moduli and static and dynamic signals of microcantilever were mainly discussed in the state of indentation compression. In addition, there are some interesting work in AFM stretching experiments to measure the spring constant and Chair-boat transitions of single molecule or molecular complex[20], and these interesting works will be further investigated.

(9) It is unclear which cantilever tip shape is considered in the developed model. Would be the conclusions different for cone, pyramid, cylinder, or microspheres of different radii?

Response: Thanks for this comment. We have added the discussion on cantilever tip shape in lines 209-210. Different from our previous work (the contact model, which considers the cantilever tip shape, https://doi.org/10.1007/s00707-024-04057-z), our present model characterizes the contact part between the cell and the microcantilever as a viscoelastic moving boundary condition, mainly focuses on the joint contribution of peripheral brush and intra-cellular structure. The influence of tip shape is ignored and the contact area does not change with time [21, 35]. The related problem of tip shape will be further studied.

(10) Parameter λ of the model, the contact area ratio of peripheral brush to the cellular inner region, what was it? Would it depend on the tip shape? Is it a function of the strain?

Response: Thanks for this comment. We have added the discussion on the parameter λ, which is a constant value and does not depend on the tip shape.

(11) Lines 191, 192 “γ is the dissipative coefficient of background (in air).” Why in air if all experiments with cells are performed in liquid?

Response: Thanks for this comment. We have added the necessary statements in lines 201-204. Usually the AFM-based biomolecular experiments are conducted in liquid [20], while the AFM-based cell experiments are conducted in both air and liquid [34], and the simulated environment of present model is in air. The fluid-solid coupling model in consideration of liquid effects will be studied in future.

(12) When compared with the Darling's results, could the authors obtain the same results when there is no peripheral brush in the model (reduced model)?

Response: We agree with this comment. We have added the results with no peripheral brush (reduced model) in Fig. 3b and Table. 1.

(13) Please explain Fig. 3c, why the peak is around 50 Hz? How is it related to the Darling's work?

Response: We agree with this comment. We have added the necessary statements in lines 284-288. Fig. 3c is a diagram of DQM convergence in AFM dynamic force-clamp mode, and it has no relevance to the Darling's work in AFM quasi-static contact mode.

(14)Figure 6. Too small text in subplots.

Response: We agree with this comment. We have resized the text in Figure 6.

Comments on the Quality of English Language: Minor editing of English language required.Please use conventional AFM and biological terminology

Response: Thanks for careful reading. We have checked the language carefully in manuscript, and used the conventional AFM and biological terminology.

Reviewer 2 Report

Comments and Suggestions for Authors

 In this manuscript, a piecewise linear viscoelastic constitutive model of cell is established considering the joint contribution of peripheral brush and intra-cellular structure. The signal interpretation models for quasi-static and dynamic signals of microcantilever are solved. The results reveal that the peripheral brush have significant effects on the complex modulus of cell and multi-channel signals of microcantilever, and suggest an alternative mapping method by collecting multi-channel signals including quasi-static and higher frequency signals with more brush indexes could be potentially used to identify cancerous cells.

Acceptation is recommended after minor revisions.

Please answer the following questions.

1. Line 44, three models are listed, but there are no discussions on the tapping mode in the following.

2. The information of the methods and experimental procedures is missing, please show the details of the experimental conditions, such as the name and version of the software, etc.

3. Please check the formats of the references carefully. For instance, there are some references with the start and end page numbers, while others only own the start page numbers.

Author Response

Reviewer #2

Comments:

(1) Line 44, three models are listed, but there are no discussions on the tapping mode in the following.

Response: We agree with this comment. Combined with reviewer 1's comment, we have rewritten the contents of lines 42-53 in manuscript.

(2) The information of the methods and experimental procedures is missing, please show the details of the experimental conditions, such as the name and version of the software, etc.

Response: Thanks for this comment. We didn't do experiment in the manuscript, the present analytical result is compared with the relevant experimental results. We have added the necessary experimental conditions in lines 272-290 and changed the “Solution Procedure” to “Solution Methods” in line 216.

(3) Please check the formats of the references carefully. For instance, there are some references with the start and end page numbers, while others only own the start page numbers.

Response: Thanks for careful reading. We have checked the formats of references and language carefully.

Reviewer 3 Report

Comments and Suggestions for Authors

The manuscript reports a theoretical model to deduce the mechanical properties of cells from AFM-base experiments. Theoretical expressions are deduced for both quasi-static and oscillatory AFM methods. The viscoelastic model is based on a SLS model which is coupled in series with a linear spring. This viscoelastic structural model aims to describe AFM experiments performed on living cells. In particular, the linear spring is included to describe the properties of the so called ‘peripheral brush of the cell’.

The manuscript might be published after some revision.  The manuscript should provide a definition of the cell peripheral brush layer in terms of well-defined biological structures.  

Author Response

Comments:

The manuscript should provide a definition of the cell peripheral brush layer in terms of well-defined biological structures.  

Response: We agree with this comment. Combined with reviewer 1's comment, we have added the definition of the cell peripheral brush layer (consist mainly of microvilli, microridges and cilia, as seen in Ref. [1] in manuscript) in lines 31-32.

Round 2

Reviewer 1 Report

Comments and Suggestions for Authors

The paper has improved, and the authors have answered some of my comments, but not sufficiently yet. 

1)  Lines 29-30: “Cellular brush is one of the important peripheral microstructures of living cell, which consist mainly of microvilli, microridges and cilia [1].” I still believe that the cellular brush that the authors consider in the study is the molecular brush, composed of the glycocalyx layer and the pericellular molecular coating. Because neither microvilli nor microridges and cilia have enough density for applying continuous models for AFM-based experiments (see any images of the microvilli on the cell surface, SEM or AFM from the cited works). I recommend authors discuss this matter in the text.

2)  It is still unclear what the dynamic force-clamp mode of AFM is. Please describe in more detail how it is performed in experiments, stage-by-stage description preferably (at least a few sentences). Mainly, it is unclear how the amplitude-frequency curve is acquired from the experiment and from which experiment. What is a “double-peak resonance mode of microcantilever” also requires a much clearer description.

3)  I strongly recommend making a section about the limitations of the current model and future developments. Please sum up the limitations (tension/compression, tip shape and contact area, parameters in air/in liquid).

4)  It is very strange that the authors did not get the same results as in the work of Darling (the results with no peripheral brush (reduced model) in Fig. 3b and Table. 1). What could be a reason for that? Please discuss in the text.

Comments on the Quality of English Language

Minor editing of English language required for grammar and style.

Author Response

Comments:

(1) Lines 29-30: “Cellular brush is one of the important peripheral microstructures of living cell, which consist mainly of microvilli, microridges and cilia [1].” I still believe that the cellular brush that the authors consider in the study is the molecular brush, composed of the glycocalyx layer and the pericellular molecular coating. Because neither microvilli nor microridges and cilia have enough density for applying continuous models for AFM-based experiments (see any images of the microvilli on the cell surface, SEM or AFM from the cited works). I recommend authors discuss this matter in the text.

Response: We agree with this comment. We have added the discussion on cellular brush in lines 121-134, 165-166, and improved Fig. 2. The cell surface structure is very complex, and the cell surface structures that have been recognized so far are: glycocalyx layer, pericellular molecular coating, microvilli, microridges and cilia, etc. [27, DOI 10.1016/j.cell.2007.04.035, https: //doi.org/10.1038/ncb1007-1110]. In the literatures [1, 27, 30] and our manuscript, the cellular peripheral brush consists of microvilli, microridges, cilia, glycosaccharide, glycoprotein (glycocalyx), etc, not only consists of the glycocalyx layer and pericellular molecular coating. In some SEM and EM scanning experiments, there were a lot of microvilli on the cell surface (as shown in Figures s1-s2 below), which was regarded as a columnar structure, and included the microfilaments (as shown in Figure s3 below). The molecules, microvilli, microridges and cilia are usually characterized by the single/double brush model, elastic/viscoelastic model, freely jointed chain model (FJC), and worm chain model (WLC) [1,20,17,28]. It should be pointed out that, most current models of cellular peripheral brush seem not well-established. In the manuscript, the peripheral brush is coarsely grained into a given number of springs (disperse model), and the single brush is assumed as a nonlinear elastic spring, while the intra-cellular structure is simplified as a continuous viscoelastic cylinder (standard linear solid model) [21,30,32].

(2)  It is still unclear what the dynamic force-clamp mode of AFM is. Please describe in more detail how it is performed in experiments, stage-by-stage description preferably (at least a few sentences). Mainly, it is unclear how the amplitude-frequency curve is acquired from the experiment and from which experiment. What is a “double-peak resonance mode of microcantilever” also requires a much clearer description.

Response: We agree with this comment. We have added the description of AFM dynamic force-clamp mode in lines 96-104, and 374-384. The cell is strained or compressed by pulling piezo driver of microcantilever. At the same time, a small-amplitude dithering is applied by the dithering piezo driver of microcantilever with low-driving frequency (i.e. 40−240 Hz). The signals of tip-sample interaction force (static force signal), amplitude and phase are collected [20]. In AFM dynamic force-clamp experiment, with the movement of pulling piezo driver (v=25 nm/s) and a small periodic excitation in the dithering piezo driver (A=10 nm, the frequency f=100 Hz), the dynamic signals of the bending deflection-time curves of microcantilever are obtained, as shown in the inset of Figure 6 a-d. In Figure 6e-h, by using the fast Fourier transform method, the amplitude-frequency curve is generated by the dynamic bending deflection-time curve of microcantilever. The two peaks (i.e., double-peak resonance) are seen in the amplitude-frequency curve of the microcantilever.

(3) I strongly recommend making a section about the limitations of the current model and future developments. Please sum up the limitations (tension/compression, tip shape and contact area, parameters in air/in liquid).

Response: We agree with this comment. We have added a section (section 4.5) to discuss the limitations of the current model and future developments in lines 425-435. The present model is mainly focuses on the joint contribution of peripheral brush and intra-cellular structure. The effects of peripheral brush on the complex moduli and static and dynamic signals of microcantilever are mainly discussed in the state of indentation compression in air. The influence of tip shape is ignored and the contact area does not change with time. In future, more completed model for cellular peripheral brush will be further investigated, as well as the fluid-solid coupling model in consideration of liquid effects and tip shape of microcantilever.

(4) It is very strange that the authors did not get the same results as in the work of Darling (the results with no peripheral brush (reduced model) in Fig. 3b and Table. 1). What could be a reason for that? Please discuss in the text.

Response: We agree with this comment. We have made a more accurate description of the relevant content in lines 295-300. Due to the different theories and cell constitutive models, the mechanical parameters fitted by the present model and reduced model (no brush) [21] are significantly different from those obtained by the Darling's model. (Darling's model: Hertz model and SLS model [30]; our reduced model: continuum mechanics model and SLS model [21]).

Minor editing of English language required for grammar and style.

Response: Thanks for careful reading. We have checked the English language carefully.
